# Epidemiological and molecular evidence of intrafamilial transmission through sexual and vertical routes in Bahia, the state with the highest prevalence of HTLV-1 in Brazil

**Aidê Nunes da Silva**[1⊙], **Thessika Hialla Almeida Araújo**[1⊙], **Ney Boa-Sorte**[1], **Giovanne Farias**[1], **Ana Karina Galvão-Barroso**[1], **Antônio de Carvalho**[1], **Ana Carolina Vicente**[2], **Bernardo Galvão-Castro**[1,3]*, **Maria Fernanda Rios Grassi**[1,3]*

1 Escola Bahiana de Medicina e Saúde Pública, Salvador, Brasil, 2 Instituto Oswaldo Cruz, Fundação Oswaldo Cruz, Rio de Janeiro, Brasil, 3 Laboratório Avançado de Saúde Pública, Instituto Gonçalo Moniz, Fundação Oswaldo Cruz, Salvador, Brasil

⊙ These authors contributed equally to this work.
* bgalvao@bahiana.edu.br (BGC); fernanda.grassi@fiocruz.br (MFRG)

**Data Availability Statement:** All relevant data are included in the paper.

## Abstract

### Introduction

Familial clustering of HTLV-1 and related diseases has been reported in Brazil. However, intrafamilial transmission of HTLV-1 based on molecular analysis has been studied only in a few communities of Japanese immigrants and African-Brazilians.

### Objective

To investigate the familial clustering of HTLV-1 infection and to determine the likely routes of transmission through epidemiological and genetic analyzes.

### Methods

Medical records of 1,759 HTLV-1+ patients from de the Center for HTLV in Salvador, Brazil, were evaluated to identify first-degree relatives previously tested for HTLV-1. Familial clustering was assumed if more than one member of the same family was HTLV-1+. LTR regions of HTLV-1 sequences were analyzed for the presence of intrafamilial polymorphisms. Family pedigrees were constructed and analyzed to infer the likely transmission routes of HTLV-1.

### Results

In 154 patients at least one other family member had tested positive for HTLV-1 (a total of 182 first-degree relatives). Of the 91 couples (182 individuals), 51.6% were breastfed, and 67.4% reported never using a condom. Of the 42 mother-child pairs, 23.8% had a child aged 13 years or younger; all mothers reported breastfeeding their babies. Pedigrees of families with 4 or more members suggests that vertical transmission is a likely mode of transmission in three families. Three families may have had both vertical and sexual transmission routes

**Funding:** This work was supported by the Coordination of Superior Level Staff Improvement-Brazil (CAPES) - Finance Code 001 (to MFRG) and the National Foundation for the Development of Private Higher Education (FUNADESP), Grants 9600140 (to MFRG) and 9600141 (to BGC). MFRG and BG-C are research fellows of CNPq (Process Nos. 308167/2021-0 and 473667/2012-6, respectively). The funders had no role in study design, data collection and analysis, decision to publish, or preparation of the manuscript.

**Competing interests:** The authors declare that the research was conducted in the absence of any commercial or financial relationships that could be construed as a potential conflict of interest.

for HTLV-1. The genetic signatures of the LTR region of 8 families revealed 3 families with evidence of vertical transmission, another 3 families (spouses) with sexual transmission, and one family with both transmission routes. HTLV-1 sequences belonged to Cosmopolitan subtype HTLV-1a Transcontinental subgroup A.

## Conclusion

Sexual and vertical transmission routes contribute to the intrafamilial spread of HTLV-1 in the state of Bahia.

## Author summary

Human T-lymphotropic virus type 1 (HTLV-1) was the first human retrovirus isolated in the early 1980s. It is estimated that approximately 10 million people worldwide are currently infected with HTLV-1, and most people living with HTLV (PLwHTLV) live in developing countries. The virus is associated with a wide range of diseases, including neoplasms such as adult T-cell leukemia/lymphoma and progressive and disabling myelopathy, but most PLwHTLV are unaware of their serologic status. HTLV-1 is transmitted through contact with contaminated blood and derivatives, sexually, and from mother to child, especially through breastfeeding. Only recently has WHO recognized HTLV-1 as a as threatening pathogen to human, but in many parts of the world HTLV screening is not performed in blood banks or in pregnant women. This may promote silent intrafamilial transmission of the virus across generations and promote familial clustering of the virus and associated diseases. In this study, we investigated the familial clustering of HTLV-1 infection in the state of Bahia, an endemic area for this virus in Brazil. We found that both sexual and vertical pathways contribute to the transmission and persistence of the virus in families across multiple generations. Therefore, in addition to expanding screening for pregnant women and providing infant formula to infected mothers, it is of utmost importance to combat sexual transmission through effective measures that can help address this serious and neglected public health problem.

## Introduction

Human T-lymphotropic virus type 1 was the first human retrovirus to be described [1]. HTLV-2, which is rarely associated with disease, was then isolated soon after [2]. HTLV-1 is the causative agent of adult T-cell leukemia/lymphoma (ATLL) [3], tropical spastic paraparesis/ HTLV-1-associated myelopathy (HAM/TSP) [4,5], HTLV-1-associated uveitis (HAU) [6] and HTLV-1-associated infectious dermatitis (HAID) [7]. In addition, many other diseases have been associated with HTLV-1 infection, including polymyositis, sinusitis, bronchoalveolar pneumonia, keratoconjunctivitis sicca and bronchiectasis, suggesting multisystemic involvement [8–10].

The worldwide prevalence of HTLV-1 infection is estimated to be between 5 and 10 million cases [8]. Of all affected countries, Brazil may be the one with the highest absolute number of persons living with HTLV-1 (PLwHTLV) [8,11]. Salvador, the capital of the state of Bahia, located in northeastern Brazil, has the highest prevalence in the country [12]. A representative sampling of this city's general population revealed a seroprevalence of 1.8%, corresponding to around 50,000 HTLV-infected individuals [13]. An ecological prevalence study published in

2019 reported that HTLV-1 is widespread throughout the state of Bahia, and that at least 130,000 individuals were infected with the virus [14].

Horizontal transmission through sexual contact appears to be the main route of HTLV-1 infection in the general population of Salvador [15]. However, there is evidence that vertical transmission also occurs in the state of Bahia, as the prevalence of HTLV-1 infection in pregnant women has been consistently estimated at around 1% [16–18]; moreover, approximately 2% of HTLV-1-infected individuals are reportedly aged 15 years or younger [14].

The prevalence of HTLV-1 is heterogenous throughout the country, with higher prevalence noted in some geographic regions, especially northeastern Brazil [19]. HTLV-1 infection tends to be more predominant with age, with females being more affected [8]. The familial aggregation of HTLV-1 infection, as well as its associated diseases, has been reported worldwide, including in Brazil [18,20–23]. However, in this country, the intrafamilial transmission of HTLV-1 based on molecular analysis has only been studied in communities of Japanese immigrants and their descendants [24–26], and in isolated communities of African descent [27].

The performance of viral genetic analysis to identify intrafamilial polymorphisms, i.e., variations in the genetic sequence of a virus within a particular family, can provide insight into the mode of HTLV-1 transmission. While the evolutionary rate of HTLV-1 is quite low, the long terminal repeat (LTR) and envelope (env) regions of the virus exhibit higher variability compared with other genomic regions [28,29]. This variability serves as a relevant molecular marker for monitoring transmission pathways and can be used to distinguish between sexual and vertical transmission [25,30,31]. The present study aimed to investigate the familial clustering of HTLV-1 infection in the state of Bahia, as well as to determine likely routes of transmission through epidemiological and genetic analyses.

## Methods

### Ethics statement

The study was approved by the Institutional Research Board of the Escola Bahiana de Medicina e Saúde Pública (CAAE: 60554416.8.0000.5544), and informed written consent was obtained when possible. In accordance with the ethical principles outlined in Brazilian regulations, the need for consent was waived only in the case of death or loss to follow-up. For child participants, formal consent was given from the parent/guardian.

### Study design and population

The present cross-sectional study was conducted between February 2017 and November 2021 at the Integrative and Multidisciplinary Center for HTLV (CHTLV), Bahia School of Medicine and Public Health Salvador (EBMSP), in Salvador, Bahia-Brazil. This outpatient clinic provides comprehensive biopsychosocial care to the public and is supported by the public health services of the Brazilian National Health System (Sistema Único de Saúde [SUS]) [32].

The medical records of 1,759 individuals were evaluated to identify first-degree relatives (parents, children, siblings and spouses) who had previously been tested for HTLV-1 by ELISA, with confirmation by Western blotting performed by CHTLV and/or the Central Laboratory of Bahia (LACEN), the reference institution for infectious diseases in the state of Bahia (Fig 1). Families with at least two members (including the patient) who had been tested were included in the study. Patients and family members with cognitive impairment, psychiatric/psychological disorders, and those diagnosed with HTLV-2 and/or coinfected with HIV, HBV or HCV were excluded.

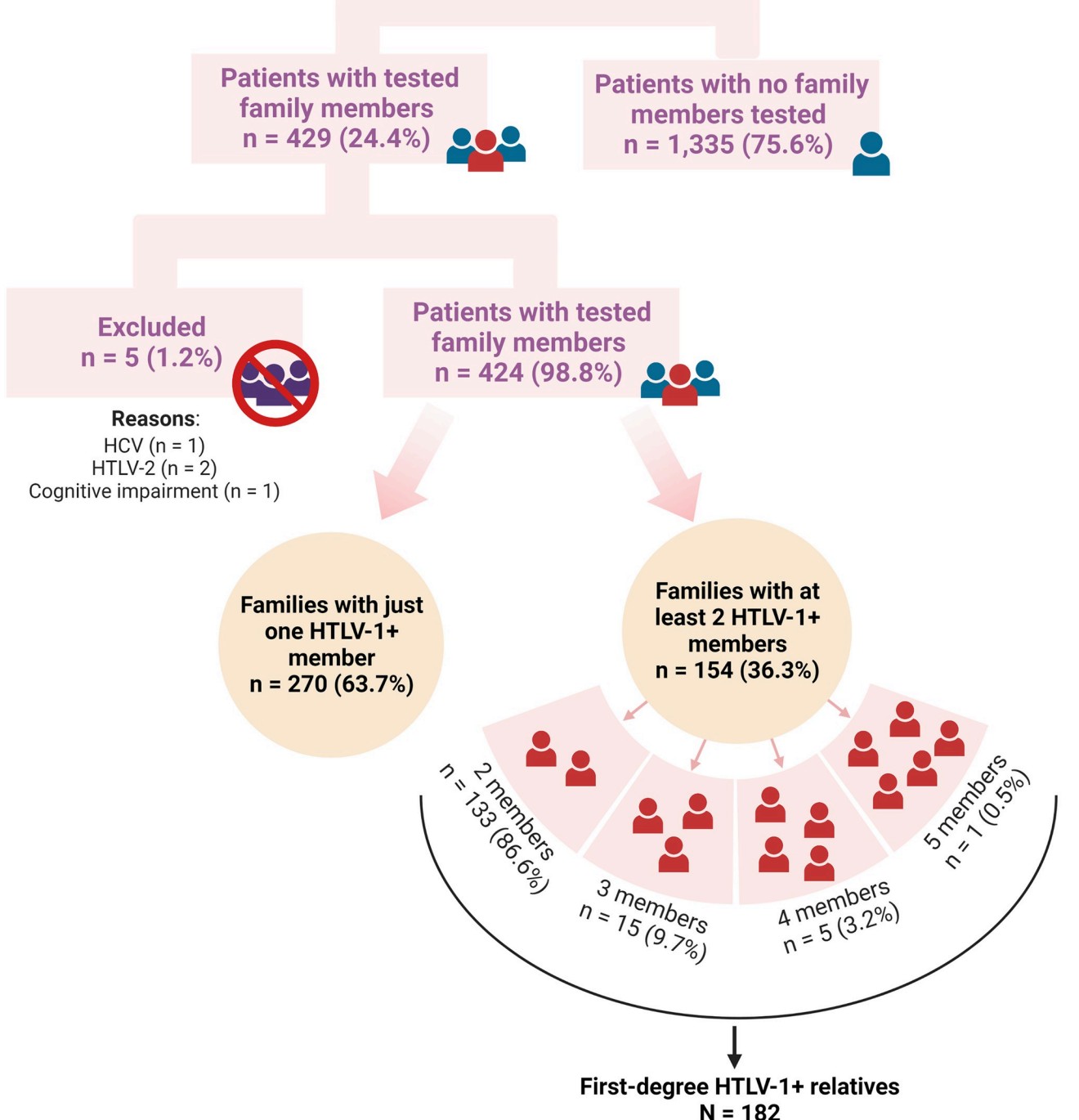

**Fig 1. Study design flowchart.** Created with BioRender.com.

### Genetic sequencing

Previously obtained DNA samples from 20 individuals (eight index cases and 12 HTLV-1 sero-positive family members) were used for genotypic analysis of the HTLV-1 LTR region. For these cases, up to three generations (grandparents, parents and grandchildren) of family members were evaluated for parenteral exposure, sexual and vertical risk factors for HTLV-1 infection, using secondary information on other family members provided by the index case.

DNA was extracted using a QIAamp DNA Blood Mini Kit (Qiagen, Hilden, Germany) in accordance with the manufacturer's instructions, and then stored at -20˚C until use. The HTLV-1 long terminal repeat (LTR) region was amplified by nested PCR using a previously described protocol [33]. PCR products were purified using a QIAamp PCR purification kit (Qiagen). Direct nucleotide sequencing was performed in both directions using a Big Dye Terminator v 3.1 Cycle Sequencing Ready Reaction Kit (Applied Biosystems, Foster City, CA) on a 3100 Automated DNA Sequencer (Applied Biosystems). Subsequently, the LTR sequences were uploaded to the HTLV-1 and 2 Genotyping Tool to identify HTLV-1 subtype [34]. All HTLV-1 LTR sequences were edited and aligned using BioEdit v5.0.9 software (Department of Microbiology, North Carolina State University, USA) and then visually analyzed to detect the presence of intrafamilial polymorphisms. The new nucleotide sequences identified in the present study were deposited in the GenBank database under accession numbers OP831963-OP831989.

### Evaluation of HTLV-1 familial aggregation

Familial aggregation was considered when more than one member of the same family tested positive for HTLV-1, with the index case being the first family member registered at CHTLV. Semi-structured questionnaires were used to collect sociodemographic, clinical, and behavioral data. Patients with HAM/TSP were diagnosed according to previously described criteria [35]. To determine the likely route of HTLV-1 infection, epidemiologic data and risk factors from family members were evaluated in combination with available LTR sequence signatures. Identical signatures between index cases in the same family were considered as evidence of vertical or sexual transmission, depending on the degree of relatedness (mother and siblings, or wife and husband).

### Data analysis

Descriptive statistics are reported as means ± SD for continuous variables, or frequencies (%) for categorical variables. Student's t test and chi-square test or Fisher's exact test were used for comparisons. Family pedigrees were generated and analyzed to infer likely routes of HTLV-1 transmission between family members. All data were analyzed using SPSS v20 software (SPSS Inc, Chicago, USA). Statistical significance was assumed when $p < 0.05$.

## Results

We identified 424 HTLV-1+ patients with at least one first-degree relative previously serologically tested for HTLV-1; of these, 154 patients (index cases) had at least one other family member who was positive for HTLV-1. In all, 182 first-degree relatives of index cases tested positive for HTLV-1 (Fig 1). The prevalence of HTLV-1 in family members was estimated at 32.9% (182/553 tested family members). The overall prevalence of familial clustering of HTLV-1 was 36.3% (154/424; CI95%:31.7–41.1). The mean ages (SD) of the index cases and their family members were 44.2 (16.12) years and 41.1 (18.64) years, respectively (p = 0.781). Females were more prevalent among both index cases and family members (p = 0.009). Most individuals

**Table 1. Composition of families of HTLV-1-infected patients (index cases).**

| Family composition | No. of families | No. of individuals |
|---|---|---|
| **Two-member families** | **133** | **266** |
| IC + spouse | 91 | 182 |
| Mother + child | 42 | 84 |
| **Three-member families** | **15** | **45** |
| IC + current wife + ex-wife | 2 | 6 |
| IC + husband + son/daughter | 7 | 21 |
| IC + sons | 2 | 6 |
| IC + husband + mother | 1 | 3 |
| IC + mother + father | 1 | 3 |
| IC + mother + son | 1 | 3 |
| **Four-member families** | **5** | **20** |
| IC + husband + 2 children | 1 | 4 |
| IC + 3 children | 2 | 8 |
| IC + wife + 2 children | 1 | 4 |
| IC + mother + 2 children | 1 | 4 |
| **Five-member families** | **1** | **5** |
| IC + mother + father + son + daughter | 1 | 5 |
| **Total** | **154** | **336** |

IC: Index case

self-reported brown or black skin color. Many family members, 40.9% (63/154) of index cases and 39% (71/182) of first-degree relatives, had less than seven years of formal schooling (Table 1). The prevalence of HAM/TSP was higher in index cases (61/154; 39.6%) compared to infected family members (37/182; 20.3%) (p < 0.001).

In 133/154 families with HTLV-1 aggregation, two members were found to be infected: 91 (68.4%) were husband and wife, while 42 (31.6%) were mother and child. In addition, there were three infected persons in 15 families, while five families had four HTLV-1+ members, and in one family five individuals tested positive (Fig 1, Table 2).

Of the 91 HTLV-1+ couples (spouses), totaling 182 individuals, 51.6% (94/182) reported being breastfed, and the majority (67.4%; 122/182) reported never using a condom. Of the 42 mother-child pairs, 10 (23.8%) children were aged 13 years or younger, all mothers reported breastfeeding their babies, and none of the mothers, nor their children, had received blood transfusions. The evaluation of pedigrees of families with three or more HTLV-1+ members suggests the likelihood of vertical transmission in seven families (#22, #30, #36, #77, #87 #101, #124) (Fig 2), sexual transmission in two families (#2, #4) (Fig 2), while both vertical and sexual routes of HTLV-1 transmission may have occurred in eleven families (#9, #32, #65, #80, #97, #114, #120, #140, #142, #144, #146) (Fig 2).

In eight families, in addition to epidemiological data, it was possible to perform genetic sequencing of the LTR region to better assess the transmission routes of HTLV-1 (Fig 3). The index case of family #152 is a mother whose molecular LTR signature (*gatttaac*) differed from that of her daughter (*aattcgat*). She was diagnosed with HTLV-1 at age 70 during a routine examination. She reported being breastfed and had only two lifetime sexual partners (both with unknown HTLV serology). She received a blood transfusion in 1989, before HTLV-1 screening was mandatory for blood donors in Brazil [36]. All five of her children, who were all born before she had received the blood transfusion, were breastfed for 6 months. Only one

**Table 2. Sociodemographic and clinical profile of index cases and family members with HTLV-1.**

| Variable | Index case N (%) 154 | Family members N (%) 182 | p value* |
|---|---|---|---|
| Sex | | | 0.009 |
| Male | 47 (30.5) | 81(44.5) | |
| Female | 107(69.5) | 101(55.5) | |
| Age (years) | | | 0.059 |
| 0 to 19 | 9 (5.84) | 26 (14.3) | |
| 20 to 39 | 51 (33.11) | 58 (31.8) | |
| 40 to 59 | 71 (46.1) | 68 (37.3) | |
| 60 to 79 | 20 (12.9) | 28 (15.3) | |
| 79+ | 3 (2.0) | 2 (1.09) | |
| Skin color (self-reported) | | | 0.542 |
| Black or mixed-race | 107 (69.4) | 121 (66.5) | |
| White | 12 (7.8) | 11 (6.0) | |
| Missing information | 35 (22.8) | 50 (27.5) | |
| Marital status | | | 0.331 |
| Single | 36 (23.3) | 47 (26.7) | |
| Married/Stable relationship | 106 (68.8) | 110 (60.4) | |
| Divorced | 4 (2.6) | 5 (2.74) | |
| Widowed | 6 (3.89) | 13 (7.14) | |
| Missing information | 2 (1.29) | 7 (3.84) | |
| Years of education | | | 0.934 |
| ≤ 7 Years | 63 (40.9) | 71 (39.0) | |
| > 7 Years | 85 (55.2) | 94 (51.7) | |
| Missing information | 6 (3.9) | 17 (9.3) | |
| HAM/TSP diagnosis | 61 (39.6) | 37 (20.3) | 0.001 |

*Chi-squared test

daughter was HTLV-1 positive by suspected sexual route of transmission. This infected daughter reported having five lifetime sexual partners and had never used a condom. Her two children were both HTLV-1 negative and were each breastfed for >6 months. The serological status of her children's father is unknown. Family #153 is composed of a mother (index case) and one of her two HTLV-1-positive daughters with an identical LTR molecular signature (*gatacggc*), likely indicating infection by vertical route. The 84-year-old mother was diagnosed with HTLV-1 20 years ago, and has HAM/TSP. She became sexually activate at age 18, had 5 lifetime sexual partners and never used a condom. She has no reported history of blood transfusion. She had 8 children who were breastfed for > 6 months: three males (only one tested negative) and five females (three tested, two with positive serology). Vertical and sexual transmission likely both played a role in the familial clustering observed in family #154. The case index (the mother) exhibited an identical LTR signature (*gatttaac*) as one of her daughters, yet another daughter had a different signature (*aattcgat*). It is important to note that the mother was married three times. Of her 13 children from the first and second marriages, 12 were tested for HTLV and only one daughter (from the first marriage) was found to be infected. As this daughter had a different signature than her mother, she likely contracted the virus through sexual transmission, since she was married to an HTLV-1-positive partner (their daughter, who was breastfed for more than 6 months, also acquired the infection). The mother's other HTLV-1-positive daughter from her third marriage was likely infected by vertical route, as she

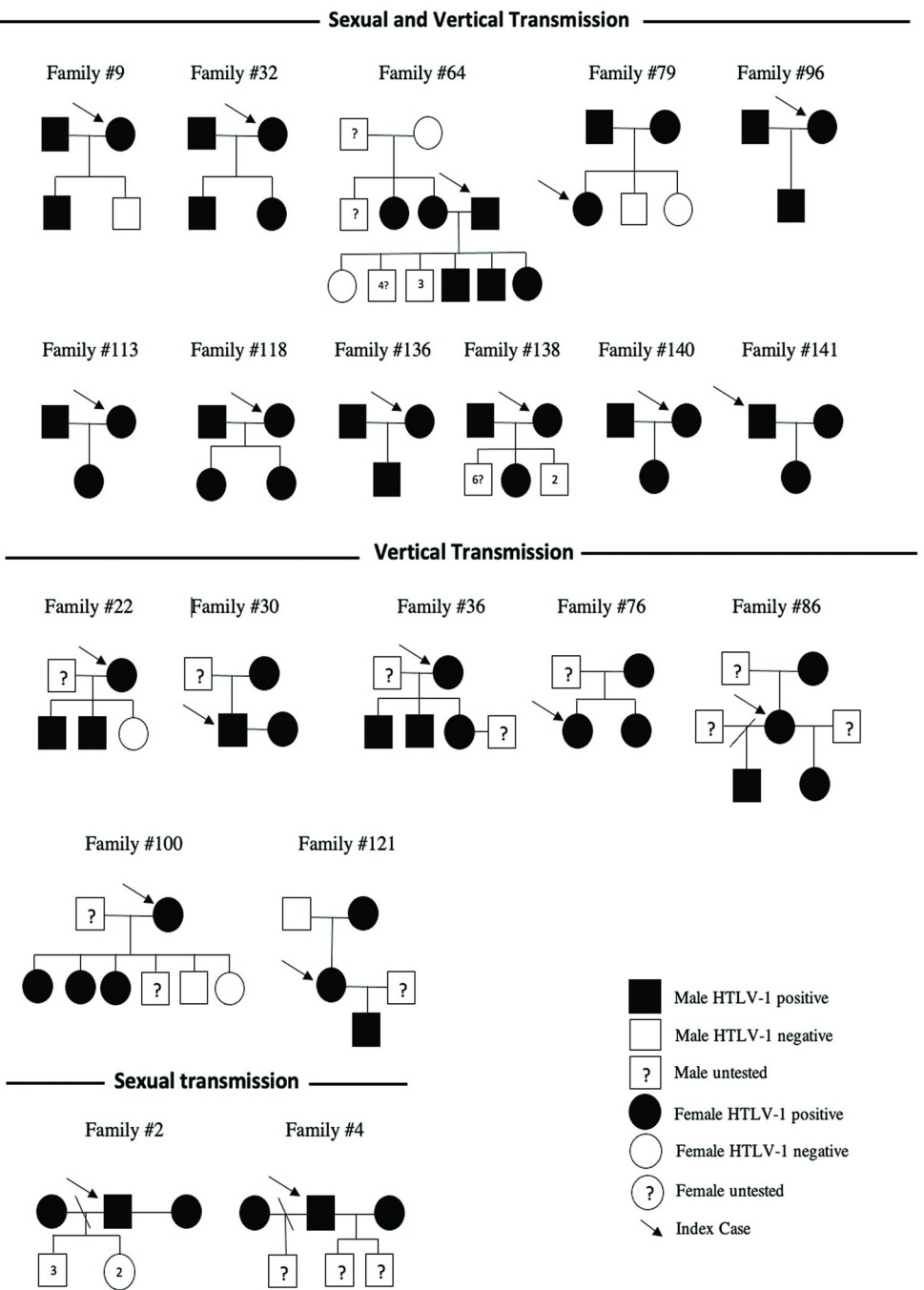

**Fig 2. Pedigrees of 20 families with likely routes of HTLV-1 transmission based on epidemiological data and sexual behavior.**

exhibited an identical LTR sequence. She was breastfed for more than 6 months, never had sexual intercourse, and never received a blood transfusion. At age 17, she was diagnosed with HAM/TSP. The likely cause of the HTLV-1 clustering observed in families #155, #156 and #157 is sexual transmission, as indicated by matching LTR signatures between index cases and their partners. In family #155 (150), the index case, the wife, was diagnosed with HAM/TSP in 2014 during routine serologic testing. She reported being breastfed, having only one sexual

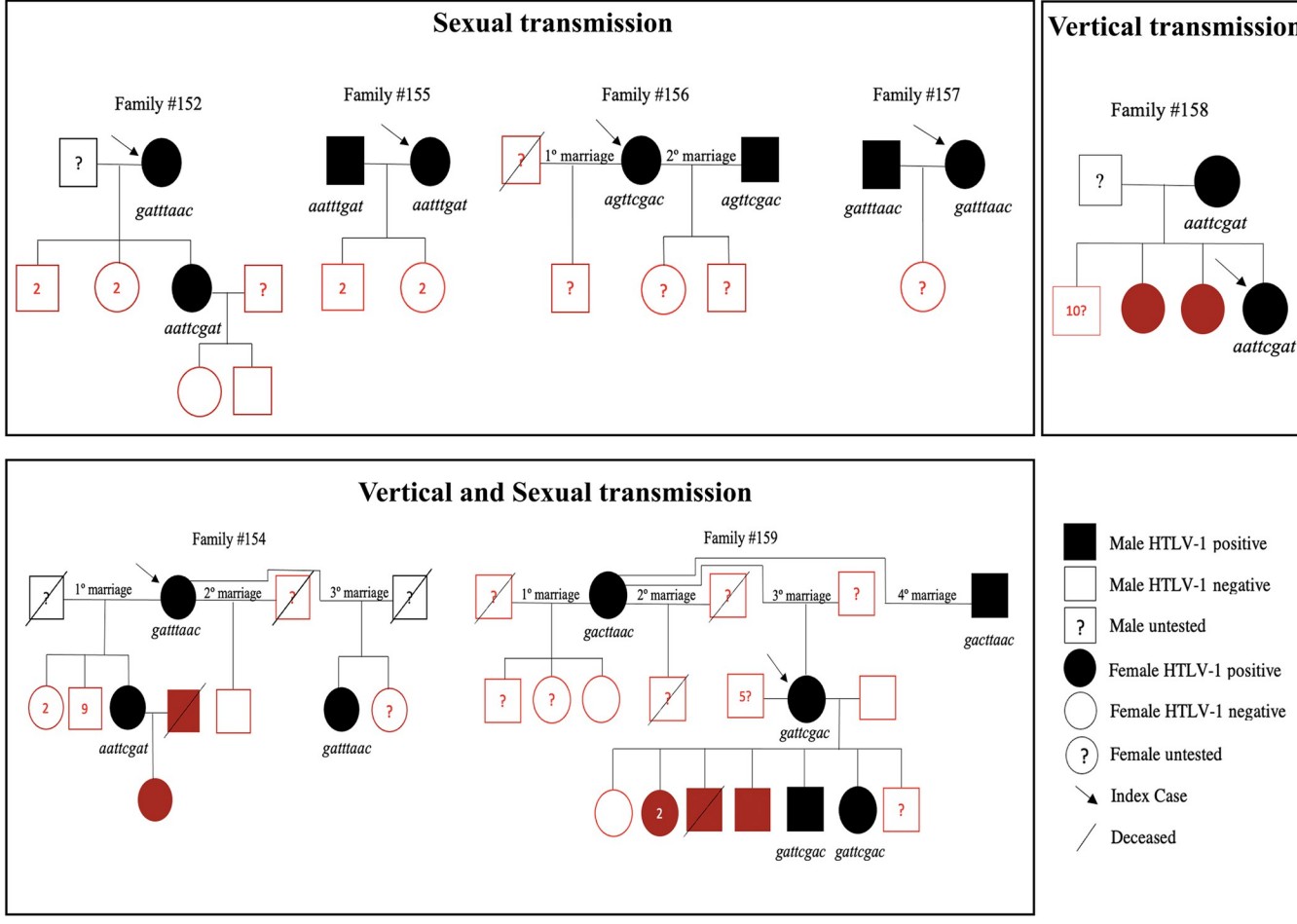

**Fig 3. Pedigrees of 8 families with likely routes of HTLV-1 transmission based on epidemiological data, sexual behavior and genetic sequencing of the HTLV-1 LTR region.** Family members highlighted in red represent, partners, children, or grandchildren of the index case who provided secondary information about sexual behavior and other risk factors.

partner, and never using a condom. Additionally, she received a blood transfusion in 1987. Her husband had a history of multiple sexual partners (> 20) and STIs, was diagnosed with HTLV infection at the same time as his wife. The HTLV-1 serological status of the couple's four children remains unknown. In family #156, the index case, the wife, was diagnosed with HTLV in 2012 when she presented neurological symptoms. She reported being breastfed, and both of her brothers tested negative for HTLV. Her first sexual intercourse occurred at the age of 15, and she had a total of 12 sexual partners without using condoms. She was married twice, had three children with unknown serology. Her second husband, diagnosed with HAM/TSP in 2016, had 15 sexual partners and no history of blood transfusion. In family #157, the index case (wife) was diagnosed with HTLV in the context of prenatal care in 2014, with HAM/TSP diagnosed in 2019. She does not have information about the serological status of her parents but reported being breastfed. She denies receiving blood transfusions. She reported having three sexual partners without using condoms. Despite her husband's diagnosis of HTLV-1 in 2016, there is no available information on associated risk factors. The serologic status of their child remains unknown. In the case of family #158, the index case (daughter) was diagnosed in 2018 with HTLV-1 during prenatal testing. Her mother subsequently tested positive for HTLV, indicating a probable vertical transmission route. The index case had a normal delivery

and did not breastfeed her baby. The 57-year-old mother has been married twice, but both of her husbands declined HTLV testing. She has a total of 13 children, all of whom were breastfed. Apart from the index case, two other daughters have also tested positive for HTLV-1. All of her sons refused to be tested. In Family #159, three generations of family members were evaluated. The index case of this family is a woman whose HTLV-1 LTR signature is identical to that of her two sons (*gattcgac*) and different from that of her mother (*gacttaac*), indicating a likely sexual route of HTLV-1 infection for the index case. She was breastfed, initiated sexual activity at the age of 14 and had six sexual partners, with the serology of five partners being unknown and the last partner having negative serology for HTLV. She received a diagnosis of HTLV-1 in 2013 when she exhibited symptoms of myelopathy. In 2018, she was diagnosed with HAM/TSP and died in 2020. Of her eight children with her last partner, seven underwent serology testing, and one refused testing. Of the six children who tested positive for HTLV-1, two underwent genetic sequencing, indicating vertical transmission as evidenced by identical LTR signatures to their mother. Two children experienced severe disseminated strongyloidiasis [37], and one of the two later died. Importantly, the last-born child with negative serology was not breastfed because the mother already knew she was infected. The mother of the index case was diagnosed with HTLV-1 in 2013 when she was already showing signs of myelopathy. She had been married four times. In the mother's fourth marriage, both she and her HTLV-1-positive husband underwent genetic sequencing, with identical LTR signature (*gacttaac*). Both of them were diagnosed with syphilis. Overall, seven unique genetic signatures were identified in the LTR sequences obtained from 20 members of eight families. Two signatures were identified in three families: *aattcgat* (families #152, #154, #158) and *gatttaac* (families #152, #154, #157). Of note, the cosmopolitan HTLV-1 subtype of transcontinental subgroup A was identified by sequencing the HTLV-1 LTR region in the samples examined from all families.

## Discussion

The present study employed epidemiological and molecular analysis in an attempt to elucidate the sexual and vertical pathways underlying the intrafamilial spread of HTLV-1 in the state of Bahia. Importantly, both vertical and horizontal transmission were concomitantly observed in some families.

The overall prevalence of HTLV-1 familial aggregation found herein was 36.3% CI95%:31.7–41.1), a rate similar to that observed in other endemic areas in countries such as Taiwan (38.8%) and Argentina (31.5%) [31,38], yet higher than that observed in Benin, West Africa (27.5%) [39].

Considering the prevalence of HTLV-1 familial aggregation throughout Brazil, the present rate (36.3%) is slightly lower than that previously reported in patients studied at an HTLV-1 center in the country's northern region (43.5%) [23]. A previous study conducted in Bahia examined 43 family members of 20 HTLV-1-infected pregnant women, and reported a prevalence of 32.6%, suggestive of familial clustering [18]. In consonance, the prevalence of HTLV-1 among family members tested herein was estimated at 32.9%. The populations served by HTLV centers possess similar sociodemographic profiles, e.g., a predominance of women with low income and educational levels compared to blood donors [40]. In contrast, a study investigating the family members of index cases recruited from blood donors in the southern region of Brazil identified a slightly lower prevalence (25.9%) [22].

Regarding the pathways of intrafamilial spread of HTLV-1 in Brazil, studies conducted in communities with a high prevalence of HTLV-1, such as those with Japanese migrants and their descendants [25,26] or Brazilians of African descent living in isolated communities [27],

suggested that both vertical and sexual transmission contribute to the maintenance of HTLV-1 spread across generations. Combining epidemiological data, sexual behavior of family members, and LTR HTLV-1 signatures, evidence indicating the likely pathways of intrafamilial spread of HTLV-1 was obtained. The HTLV-1 LTR region was selected as a potential marker to evaluate routes of viral transmission, since this is the genetic region of the virus with greatest hypervariability [28]. The fragments generated (480 base pairs on average, out of the 750 bp length of the entire LTR region) were sufficient to distinguish both inter- and intrafamilial signatures. These signatures served as genetic markers to infer transmission routes in accordance with the epidemiological and risk factors associated with different family members. For example, evidence of intrafamilial sexual transmission was found in families #155, #156, #157 and #159 (index case and husband from 4th marriage). In these families, multiple risk factors for HTLV-1 infection were identified, such as multiple sexual partners, past history of STI, and two partners with an identical signature. On the contrary, vertical HTLV-1 transmission was the likely route of intrafamilial HTLV-1 spread in families #153, #154 and #158, as the same signature was detected in both mothers and children. Interestingly, in family #154, one of the daughters (from the mother's third marriage) who presented the same signature as her mother had been breastfed for more than 6 months, had never had sexual intercourse, and had not received any blood transfusions. In contrast, the epidemiological and sexual history of another daughter (from her mother's first marriage), who presented a different LTR HTLV -1 signature than her mother, precludes the possibility of vertical transmission. The latter daughter was the only infected child (out of 12) from this marriage. As her deceased partner was confirmed to be HTLV-1-positive, she was most likely infected sexually. In family #152, although both the mother and one of her daughters were infected, the family pedigree, risk factors associated with HTLV-1 infection, and LTR signatures did not provide any evidence of vertical transmission. The mother (*gatttaac*), who had received a blood transfusion before serologic HTLV-1/2 screening of blood donors became mandatory in Brazil [36], gave birth to five children before receiving the blood transfusion. The infected daughter had multiple sexual partners, never used a condom, and her HTLV-1-LTR signature (*aattcgat*) differed from that of her mother. Finally, in family #159, it was possible to sequence the HTLV-1 LTR region in family members across three generations, which resulted in evidence of both sexual transmission (mother and fourth husband with the same HTLV-1-LTR signature) and vertical transmission (a daughter from the mother's third marriage and her children presented the same signature) (*gattcgac*). The results obtained herein support the role of using genetic signatures as markers for relevant modes of HTLV-1 transmission. However, it is important to note that none of the signatures identified were correlated with a particular route of transmission. It should be emphasized that as the HTLV-1 genome presents great stability as evidenced by a nucleotide exchange rate of about 1% every 1000 years [30,41], it therefore constitutes an excellent molecular marker for the evaluation of transmission route [30]. Moreover, the rate of genetic variation observed within this same HTLV-1 aA subtype is less than 0.5% [42].

As vertical transmission is considered the main route of intrafamilial spread of HTLV, preventive measures have been taken to address this concern. Measures implemented in Japan to control vertical HTLV-1 transmission, such as screening pregnant women in endemic areas and formula-feeding the babies of infected women, have successfully resulted in significantly decreased prevalence [43]. In Bahia, HTLV-1 testing for pregnant women became mandatory in 2011 [44] and has more recently been recommend across Brazil [45]. The World Health Organization and Pan American Health Organization have both strongly recommended mandatory testing for pregnant women in recognition of the importance of vertical transmission [46]. However, little attention has been paid to the intrafamilial spread of HTLV-1 through sexual intercourse [47]. Recently, an increase in the rate of transmission of HTLV-1 through

sexual intercourse, particularly among young people was observed in Japan [48]. Studies conducted in Salvador have identified sexual transmission as the main route responsible for HTLV-1 in the general population [13,15]; moreover, approximately 1% of pregnant women tested in Salvador were found to be infected with HTLV-1 [16,17].

The silent intrafamilial spread of infection via multiple modes of transmission within the same family over several generations highlights the neglected status of HTLV-1, as well as the social vulnerability associated with this infection. For example, family #159 lives in extreme poverty in a semi-isolated community with poor sanitary conditions [49]. The index case (diagnosed with HAM/TSP) died at the age of 42 years, had six children infected with HTLV-1, one of whom developed disseminated strongyloidiasis and later died [37]. Cases of Norwegian scabies have also been observed in this family. In addition, the silent dissemination of HTLV reflects a complex network of factors that impose limitations on access to testing, such as a lack of knowledge about HTLV among health professionals, deficient testing availability, few campaigns specifically aimed at preventing sexual transmission, and patient/family fear of social stigma. On the other hand, the introduction of public policies can interrupt the chain of HTLV-1 transmission if infected women are encouraged to stop breastfeeding and receive infant formula for their children, as observed in families #158 and #159. In addition, it was recently reported that mandatory HTLV screening and the cessation of breastfeeding are cost-effective measures in Brazil [50].

In addition, the familial aggregation of HTLV-1 has also been associated with the clustering of diseases related to the virus. A systematic review investigating HTLV-1-associated diseases reported a higher incidence of ATLL and HAM/TSP in relatives of index cases [20]. Moreover, the concomitant presence of HAM/TSP and strongyloidiasis in HTLV-1-infected mothers was reported to be associated with higher seropositivity in children [51]. The present study also identified HAM/TSP in approximately 20% of the first-degree relatives of HTLV-1+ patients (Table 2). Moreover, previous studies conducted in Salvador have additionally identified familial clusters of HAID in association with HAM/TSP [21,52]. Finally, another study demonstrated familial aggregation in HAM/TSP patients, with onset occurring a younger age [53].

The HTLV-1 subtype identified in the studied sample belongs to the Cosmopolitan subtype Transcontinental subgroup A (HTLV-1a). This subtype was also detected in previous studies of populations in Bahia, supporting the introduction of the virus during the post-Columbian period through the slavery trade [54,55]. Conversely, Japanese subgroup (a) was reported in a subset of samples from Japanese immigrants and their families, confirming two different pathways of HTLV-1 introduction in Brazil during the post-Colombian period [25,26].

Our study presents some limitations with respect to bias in sample selection, as the sample was mainly composed of adults with an average age of 50 years. In addition, the percentage of tested family members was low, with approximately 1.2 relatives tested per index case, and spouses representing the majority of family units in which familiar aggregation was observed.

In conclusion, the results presented herein demonstrate that both sexual and vertical transmission routes contribute to the intrafamilial spread of HTLV-1 in the state of Bahia.

Therefore, we strongly recommend the serological testing all family members of HTLV-1-infected individuals, with subsequent genetic evaluations aimed at better understanding the relevant factors related to the interfamilial spread of this virus. Considering that both vertical and sexual routes play a role in the spread of HTLV-1 [31], in addition to expanding the screening of pregnant women and providing infant formula to infected mothers, it is also of paramount importance to address sexual transmission by implementing effective measures, which may contribute to the control this serious and neglected public health problem that mainly affects socio-economically disadvantaged populations.

## Acknowledgments

We would like to thank Sônia Rangel and Maíara Cerqueira for administrative support and Noilson Lazaro Gonçalves for technical support. We would also like to thank Dr Fred Neves Santos for critically reading the manuscript and Andris K. Walter for critical analysis, English language revision, and manuscript copyediting assistance.

## Author Contributions

**Conceptualization:** Aidê Nunes da Silva, Bernardo Galvão-Castro, Maria Fernanda Rios Grassi.

**Data curation:** Aidê Nunes da Silva, Thessika Hialla Almeida Araújo, Ney Boa-Sorte, Giovanne Farias, Ana Karina Galvão-Barroso, Antônio de Carvalho, Ana Carolina Vicente, Bernardo Galvão-Castro, Maria Fernanda Rios Grassi.

**Formal analysis:** Aidê Nunes da Silva, Thessika Hialla Almeida Araújo, Ney Boa-Sorte, Giovanne Farias, Ana Karina Galvão-Barroso, Antônio de Carvalho, Ana Carolina Vicente, Bernardo Galvão-Castro, Maria Fernanda Rios Grassi.

**Funding acquisition:** Bernardo Galvão-Castro, Maria Fernanda Rios Grassi.

**Methodology:** Aidê Nunes da Silva, Ney Boa-Sorte, Bernardo Galvão-Castro, Maria Fernanda Rios Grassi.

**Supervision:** Bernardo Galvão-Castro.

**Validation:** Thessika Hialla Almeida Araújo, Ana Carolina Vicente, Maria Fernanda Rios Grassi.

**Writing – original draft:** Thessika Hialla Almeida Araújo, Ney Boa-Sorte, Giovanne Farias, Ana Karina Galvão-Barroso, Antônio de Carvalho, Ana Carolina Vicente, Bernardo Galvão-Castro, Maria Fernanda Rios Grassi.

**Writing – review & editing:** Aidê Nunes da Silva, Bernardo Galvão-Castro, Maria Fernanda Rios Grassi.

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
