## [Decision Letter · Decision Letter 0]

13 Mar 2023

Dear Dra. Rios Grassi,

Thank you very much for submitting your manuscript "Epidemiological and molecular evidence of intrafamilial transmission through sexual and vertical routes in Bahia, the state with the highest prevalence of HTLV-1 in Brazil" for consideration at PLOS Neglected Tropical Diseases. As with all papers reviewed by the journal, your manuscript was reviewed by members of the editorial board and by several independent reviewers. In light of the reviews (below this email), we would like to invite the resubmission of a significantly-revised version that takes into account the reviewers' comments. 

Dear Authors,

please see below reviewers' comments.

One of the main comments is about the limitation of the length of sequences. 

Thank you in advance.

We cannot make any decision about publication until we have seen the revised manuscript and your response to the reviewers' comments. Your revised manuscript is also likely to be sent to reviewers for further evaluation.

Sincerely,

Elvina Viennet, PhD

Section Editor

Elvina Viennet

Section Editor

Dear Authors,

please see below reviewers' comments.

One of the main comments is about the limitation of the length of sequences. 

Thank you in advance.

Reviewer's Responses to Questions

**Key Review Criteria Required for Acceptance?**

**Methods**

-Are the objectives of the study clearly articulated with a clear testable hypothesis stated?

-Is the study design appropriate to address the stated objectives?

-Is the population clearly described and appropriate for the hypothesis being tested?

-Is the sample size sufficient to ensure adequate power to address the hypothesis being tested?

-Were correct statistical analysis used to support conclusions?

-Are there concerns about ethical or regulatory requirements being met?

Reviewer #1: - Sequences are too short to be informative. Authors need to gnerate full LTR sequence, and it would even be better to have a complete genome or another gene (env?).

Reviewer #2: Methods

-Are the objectives of the study clearly articulated with a clear testable hypothesis stated?

The aims are stated in the Introduction (lines 94-96): "to investigate familial clustering of HTLV-1 infection in the state of Bahia, as well as to determine likely routes of transmission through epidemiological and genetic analysis". Please could the authors provide more information on the rationale for the use of viral genetic analysis to investigate familial clustering of HTLV-1, including defining the term 'intrafamilial polymorphisms' and how these could provide evidence of vertical versus sexual transmission. 

-Is the study design appropriate to address the stated objectives?

Yes

-Is the population clearly described and appropriate for the hypothesis being tested?

Please could some further details on the population be added. In particular, were family members tested through CHTLV or other clinics? Do we know how many family members were offered a test and declined?

-Is the sample size sufficient to ensure adequate power to address the hypothesis being tested?

Sample size calculation is not provided. Simple descriptive statistics are included and are sufficient.

-Were correct statistical analysis used to support conclusions?

Yes

-Are there concerns about ethical or regulatory requirements being met? 

The authors mention (line 268) that written informed consent was not required according to national and local specifications. Was oral consent required and if so, obtained, eg for collection of blood samples for HTLV sequencing, or completion of semi-structured questionnaires?

Other aspects 

-Spouses were identified, who had been tested for HTLV, but not unmarried partners. Could the latter also be included?

-Is the proportion of medical records lacking adequate documentation on first-degree relative HTLV testing available? 

-Why did the authors exclude patients and family members with cognitive impairment, psychiatric/psychological disorders and those with HIV/HBV/HCV? The rationale should be stated. For example, many individuals with HTLV-1 may experience psychological disorders such as depression, and excluding them may introduce bias.

-Questionnaires: were these completed by patients or healthcare professionals? If the latter, please state the sources used to complete them.

-Genetic sequencing: why were only a subset of individuals included? Were these fresh or stored samples? Presumably they were whole blood specimens, which should be stated. Please also state the rationale for choosing LTR rather than another viral gene region or whole genome sequencing.

**Results**

-Does the analysis presented match the analysis plan?

-Are the results clearly and completely presented?

-Are the figures (Tables, Images) of sufficient quality for clarity?

Reviewer #1: - AUthors have looked at the propagation of HTLV-1 within familial clusters in Bahia, Brazil. THey have studied family clusters, aroung index cases that have likely been found positive through HAM/TSP diagnosis.

They sequenced part of the LTR as a marker of transmission. The segment may be too short to be sufficiently informative.

Major question :

- What is the nucleotide difference between sequences within and between different families ?

- How many mutations per transmission chain do they report ?

- Authors should sequence the full ltr through 2 distinct PCRs and if possible generate a full genome sequence to really streghten their affirmation of intrafamilial transmission.

- what is the supposed duration of breastfeeding?

Minor point :

- Authors report that most participants do not use condoms. Could they ask for the number of previous partners.

Reviewer #2: -Does the analysis presented match the analysis plan?

Yes

-Are the results clearly and completely presented?

The genetic results need further explanation within the text ie how they were interpreted to assign sexual or vertical transmission routes.

-Are the figures (Tables, Images) of sufficient quality for clarity?

Figure 1 - please add % as well as total numbers. The authors should later discuss why the family members of most patients (1335/1759) had not been tested for HTLV.

Figure 3 - show transmission route categories of vertical / sexual / both routes for the different families

Table 1 - for widowed patients, are the deaths of any partners considered likely to be HTLV related?

Table 2 - add ATLL diagnoses

General - are proviral load data available? It would be interesting to note proviral load in patients with and without HTLV-1 positive first degree relatives, for example.

General - the authors mention reporting of condom use, breastfeeding and receipt of blood transfusions. Are any of the following data also available? Duration of breastfeeding, history of injection drug use, history of organ transplant

**Conclusions**

-Are the conclusions supported by the data presented?

-Are the limitations of analysis clearly described?

-Do the authors discuss how these data can be helpful to advance our understanding of the topic under study?

-Is public health relevance addressed?

Reviewer #1: - Data are quite weak and more information on the virus genome are required. 

- Authors should discuss their data according to other publications of intrafamilial transmission: - Other papers have looked at intrafamilial transmission and should be discussed and presented : Tuppin et al 1996 ; VanDooren et al, 2007

Reviewer #2: -Are the conclusions supported by the data presented?

Yes

-Are the limitations of analysis clearly described?

Further details should be added eg speculating on why the % of tested family members was low, and how this could be improved. Also, any limitations around how they concluded vertical versus sexual transmissions within families.

-Do the authors discuss how these data can be helpful to advance our understanding of the topic under study?

There is some discussion but this could be improved, for example, considering current barriers to family member testing; discussing in more detail the interventions which may reduce sexual transmission of HTLV-1 (clinician/patient awareness, increasing testing in STI clinics, promotion of condom use etc)

-Is public health relevance addressed?

Yes

Other points

-Line 218 - clarify which measures in Japan were implemented

-Line 221 - since the introduction of mandatory HTLV testing for pregnant women in Bahia, what has been the uptake and % positivity?

**Editorial and Data Presentation Modifications?**

Reviewer #1: (No Response)

Reviewer #2: (No Response)

**Summary and General Comments**

Reviewer #1: AUthors have looked at the propagation of HTLV-1 within familial clusters in Bahia, Brazil. THey have studied family clusters, aroung index cases that have likely been found positive through HAM/TSP diagnosis.

They sequenced part of the LTR as a marker of transmission. The segment may be too short to be sufficiently informative.

Major question :

- What is the nucleotide difference between sequences within and between different families ?

- How many mutations per transmission chain do they report ?

- Authors should sequence the full ltr through 2 distinct PCRs and if possible generate a full genome sequence to really streghten their affirmation of intrafamilial transmission.

- Other papers have looked at intrafamilial transmission and should be discussed and presented : Tuppin et al 1996 ; VanDooren et al, 2007

Minor point :

- Authors report that most participants do not use condoms. Could they ask for the number of previous partners.

Reviewer #2: The report describes the extent of familial clustering of HTLV-1 in Salvador, Bahia. Through epidemiological and viral genetic analyses, the authors conclude that both sexual and vertical transmission routes contribute to HTLV-1 transmission. The findings in a large number of individuals with HTLV-1 and their family members are of interest, given the lack of data on relative importance of different HTLV-1 transmission routes. However, there are areas throughout the report which require strengthening. In particular, clarifications are required on the background and rationale for the use of HTLV-1 genetic analyses, and how these results were interpreted.

PLOS authors have the option to publish the peer review history of their article (what does this mean?). If published, this will include your full peer review and any attached files.

Reviewer #1: No

Reviewer #2: No
---

## [Decision Letter · Decision Letter 1]

14 Sep 2023

Dear Dra. Rios Grassi,

Thank you very much for submitting your manuscript "Epidemiological and molecular evidence of intrafamilial transmission through sexual and vertical routes in Bahia, the state with the highest prevalence of HTLV-1 in Brazil" for consideration at PLOS Neglected Tropical Diseases. As with all papers reviewed by the journal, your manuscript was reviewed by members of the editorial board and by several independent reviewers. The reviewers appreciated the attention to an important topic. Based on the reviews, we are likely to accept this manuscript for publication, providing that you modify the manuscript according to the review recommendations. 

Sincerely,

Elvina Viennet, PhD

Section Editor

Elvina Viennet

Section Editor

Reviewer's Responses to Questions

**Key Review Criteria Required for Acceptance?**

**Methods**

-Are the objectives of the study clearly articulated with a clear testable hypothesis stated?

-Is the study design appropriate to address the stated objectives?

-Is the population clearly described and appropriate for the hypothesis being tested?

-Is the sample size sufficient to ensure adequate power to address the hypothesis being tested?

-Were correct statistical analysis used to support conclusions?

-Are there concerns about ethical or regulatory requirements being met?

Reviewer #1: Although authors could not sequence fully the LTR, the new version highlights the limitations.

**Results**

-Does the analysis presented match the analysis plan?

-Are the results clearly and completely presented?

-Are the figures (Tables, Images) of sufficient quality for clarity?

Reviewer #1: new figures are clear

**Conclusions**

-Are the conclusions supported by the data presented?

-Are the limitations of analysis clearly described?

-Do the authors discuss how these data can be helpful to advance our understanding of the topic under study?

-Is public health relevance addressed?

Reviewer #1: Could the authors introduce the remark they made to query 3 into the text (I have not seen it in the final text) : It is important to note that none of the signatures identified were correlated

with a particular route of transmission.

**Editorial and Data Presentation Modifications?**

Reviewer #1: I would just ask the authors to add in the discussion a remark they made to my comments:

"It is important to note that none of the signatures identified were correlated

with a particular route of transmission."

**Summary and General Comments**

Reviewer #1: (No Response)

PLOS authors have the option to publish the peer review history of their article (what does this mean?). If published, this will include your full peer review and any attached files.

Reviewer #1: Yes: Philippe Afonso

Figure Files:

Data Requirements:

Reproducibility:

References

---

## [Editor Report · Decision Letter 2]

19 Sep 2023

Dear Dra. Rios Grassi,

We are pleased to inform you that your manuscript 'Epidemiological and molecular evidence of intrafamilial transmission through sexual and vertical routes in Bahia, the state with the highest prevalence of HTLV-1 in Brazil' has been provisionally accepted for publication in PLOS Neglected Tropical Diseases.

Best regards,

Elvina Viennet, PhD

Section Editor

Elvina Viennet

Section Editor

---

## [Editor Report · Acceptance letter]

22 Sep 2023

Dear Dra. Rios Grassi,

We are delighted to inform you that your manuscript, "Epidemiological and molecular evidence of intrafamilial transmission through sexual and vertical routes in Bahia, the state with the highest prevalence of HTLV-1 in Brazil," has been formally accepted for publication in PLOS Neglected Tropical Diseases.

Best regards,

Shaden Kamhawi

co-Editor-in-Chief

Paul Brindley

co-Editor-in-Chief
